# Sanguinarine Induces Necroptosis of HCC by Targeting PKM2 Mediated Energy Metabolism

**DOI:** 10.3390/cancers16142533

**Published:** 2024-07-13

**Authors:** Rui Kong, Nan Wang, Chunli Zhou, Yuqing Zhou, Xiaoyan Guo, Dongyan Wang, Yihai Shi, Rong Wan, Yuejuan Zheng, Jie Lu

**Affiliations:** 1Department of Gastroenterology, The Affiliated Suzhou Hospital of Nanjing Medical University, Suzhou Municipal Hospital, Gusu School, Nanjing Medical University, Suzhou 215000, China; 2Department of Gastroenterology, Shanghai Tenth People’s Hospital Affiliated to Tongji University, Tongji University School of Medicine, Shanghai 200072, China; 3Department of Gastroenterology, Shanghai General Hospital, Shanghai Jiao Tong University School of Medicine, Shanghai 200080, China; wanrong1970@163.com; 4Department of Gastroenterology, Gongli Hospital of Shanghai Pudong New Area, Shanghai 200135, China; 5The Research Center for Traditional Chinese Medicine, Shanghai Institute of Infectious Diseases and Biosecurity, Shanghai University of Traditional Chinese Medicine, Shanghai 201203, China

**Keywords:** sanguinarine, necroptosis, energy metabolism, hepatocellular carcinoma, PKM2/β-catenin axis

## Abstract

**Simple Summary:**

Here, we show that the plant-derived product, sanguinarine, inhibited aerobic glycolysis and oxidative phosphorylation; which resulted in energy alternations and necroptosis of tumor cells. Moreover, we identify the PKM2/β-catenin axis as the main target in sanguinarine treatment against HCC development.

**Abstract:**

Backgrounds: Abnormal metabolism is the hallmark of hepatocellular carcinoma. Targeting energy metabolism has become the major focus of cancer therapy. The natural product, sanguinarine, displays remarkable anti-tumor properties by disturbing energy homeostasis; however, the underlying mechanism has not yet been elucidated. Methods: The anticancer activity of sanguinarine was determined using CCK-8 and colony formation assay. Morphological changes of induced cell death were observed under electron microscopy. Necroptosis and apoptosis related markers were detected using western blotting. PKM2 was identified as the target by transcriptome sequencing. Molecular docking assay was used to evaluate the binding affinity of sanguinarine to the PKM2 molecule. Furthermore, Alb-Cre^ERT2^; PKM2^loxp/loxp^; Rosa26^RFP^ mice was used to construct the model of HCC—through the intervention of sanguinarine in vitro and in vivo—to accurately explore the regulation effect of sanguinarine on cancer energy metabolism. Results: Sanguinarine inhibited tumor proliferation, metastasis and induced two modes of cell death. Molecular docking of sanguinarine with PKM2 showed appreciable binding affinity. PKM2 kinase activity and aerobic glycolysis rate declined, and mitochondrial oxidative phosphorylation was inhibited by sanguinarine application; these changes result in energy deficits and lead to necroptosis. Additionally, sanguinarine treatment prevents the translocation of PKM2 into the nucleus and suppresses the interaction of PKM2 with β-catenin; the transcriptional activity of PKM2/β-catenin signaling and its downstream genes were decreased. Conclusions: Sanguinarine showed remarkable anti-HCC activity via regulating energy metabolism by PKM2/β-catenin signaling. On the basis of these investigations, we propose that sanguinarine might be considered as a promising compound for discovery of anti-HCC drugs.

## 1. Introduction

Abnormal metabolism is a key feature of hepatocellular carcinoma (HCC) [1], with cancer cells exhibiting heightened nutrient demands and altered metabolism to meet increased energy requirements [2]. HCC is characterized by elevated aerobic glycolysis, also known as the Warburg effect [3]. This metabolic shift is often accompanied by upregulation of pyruvate kinase M2 (PKM2) in tumor cells [4,5,6], which is a critical enzyme in the conversion of phosphoenolpyruvate to pyruvate. The balance between PKM2 dimer and tetramer forms is dependent on the energy needs of the tumor, with the tetrameric form not involved in the kinase activity of the dimer form acting as a transcription co-factor in the nucleus. The increased presence of dimeric PKM2 plays a role in regulating the expression of multiple oncogenes, promoting the aggressive behavior of HCC during tumorigenesis [7].

Mitochondria plays a crucial role in cancer progression, providing biosynthetic substrates and energy that support tumor cell proliferation and metastasis. Previous studies have suggested that, despite the presence of oxygen, mitochondria can be in a dormant state with reduced bioenergetic efficiency; leading cancer cells to primarily rely on glycolysis for energy production rather than oxidative phosphorylation [8]. However, there is also research indicating active mitochondrial oxidative respiration in tumor cells [9]. While glycolysis remains the main method of energy metabolism in cancer, levels of oxidative respiration are higher in tumor cells compared to normal cells. Abnormal expression of genes related to energy metabolism has been observed in tumor necroptosis [10], making energy metabolism a key target in current therapy for hepatocellular carcinoma.

Sanguinarine, a bioactive benzophenanthridine alkaloid extracted from the roots of Sanguinaria canadensis [11], has been shown in various studies to induce diverse forms of cell death in different types of human cancers [12,13,14]. While it is known that excessive reactive oxygen species (ROS) originating from mitochondria can lead to oxidative stress and cell death through caspase-dependent apoptosis [15], the specific mechanisms of caspase-independent cell death and the metabolic alterations induced by sanguinarine remain unclear.

During the last decade, a range of non-apoptotic cell death pathways have been described [16], opening the exciting possibility to eliminate cancer cells via alternative routes. Necroptosis is a form of regulated cell death that differs from apoptosis and traditional necrosis [17]. Cellular swelling, dilatation organelles and plasma membrane protrusions are morphological characteristics of necroptosis [18]. Tumor cell necroptosis can be found in the necrotic areas of solid tumors. Down-regulated necroptosis biomarkers, including RIP3, indicated a worse prognosis in patients with breast cancer and colorectal cancer [19]. A growing arsenal of chemotherapeutics, natural compounds, and classical necroptosis inducers have been proved to disappear tumor cells through necroptosis. Shikonin-induced necroptosis was found to bypass the resistance to cancer drugs in human leukemia cell lines [20]. Resibufogenin, a bioactive compound extracted from toad venom, has been reported to suppress the growth and metastasis of colorectal cancer by activating RIPK3 necroptosis [21]. In addition, radiotherapy against thyroid cancer (ATC) and adrenocortical cancer (ACC)—by triggering necrosome formation—can cause necroptosis [22].

Emerging evidence has indicated that RIP3 and MLKL are key regulators in the necroptotic pathway. Phosphorylation of RIP3/MLKL is known to contribute to the induction of classical necroptosis. Additionally, excessive opening of mitochondrial permeability transition pores (mPTPs) and Cyclophilin D (CypD)-mediated mitochondria depolarization are essential in necroptosis [23,24]. This mode of cell death leads to severe energy deprivation and disruption of cellular integrity. Various intracellular events—such as ROS generation, mitochondrial dysfunction, and impaired glycolysis [25]—have been shown to trigger energy deprivation and necroptosis. Furthermore, there are several chemotherapeutics and natural compounds that can activate necroptosis in cancer cells, including shikonin, RETRA, and IFN [26,27,28]. In the current study, it was demonstrated that sanguinarine-induced tumor necroptosis is a result of energy deprivation.

Consequently, we aimed to confirm the function of sanguinarine in HCC and explore the underlying molecular mechanisms behind necroptosis induction and metabolism alterations.

## 2. Materials and Methods

### 2.1. Cell Viability and Colony Formation Assay

Four different HCC cell lines HCC-LM3, SMMC-7721, Bel-7402 and a hepatoblastoma cell line HepG2 were purchased from the Chinese Academy of Sciences Committee Type Culture Collection Cell Bank (Beijing, China). Cells were seeded at a density of 5000 cells/mL in 96-well plates (100 µL medium per well) with three replicates and treated with different concentrations of sanguinarine at 0.1 μM, 0.4 μM, 0.8 μM, 1 μM, 2 μM, 4 μM, 8 μM, 10 μM for 6 h. The ranges of sanguinarine concentrations were chosen based on data reported in literature [29]. In this study, 2 μM and 4 μM sanguinarine were chosen for subsequent experiments. The cell viability ratio was determined using a cell counting kit (Yeasen, Shanghai, China) according to the instructions. For colony formation, cancer cells were treated with different concentrations of sanguinarine for 14 days; the visible colonies were counted and photographed under light microscopy.

### 2.2. Chromatin Immunoprecipitation (ChIP) Assay

Chromatin immunoprecipitation (ChIP) assays were carried out using a ChIP assay kit (Beyotime Biotechnology Corporation, Shanghai, China) according to the manufacturer’s instructions. Chromatin solutions were incubated with specific primary antibodies or control IgG. DNA immunoprecipitated with β-catenin antibody was amplified with PCR using CCND1 promoter-specific primers. PCR products were then visualized by agarose gel electrophoresis.

### 2.3. Transmission Electron Microscopy

Cell samples were treated with different concentrations of sanguinarine, and fixed with 2% osmium tetroxide. Samples were then dehydrated and embedded with pure epoxy resin. The cellular ultrastructure changes were investigated using transmission electron microscopy (JEM-1230 from JEOL, Tokyo, Japan).

### 2.4. Biomedical Analysis

Lactate acid production in cells treated with sanguinarine or a negative control was assessed using the Lactate Acid Assay kit from Nanjing Jiancheng Bioengineering Institute, Nanjing, China. The glucose uptake rate was determined with the Screen Quest™ Fluorimetric Glucose Uptake Assay Kit, following the provided instructions. Oxygen consumption rate was measured using the Extracellular Oxygen Consumption Assay Kit (ab197243; Abcam, Cambridge, UK), and the extracellular acidification rate was analyzed with the Glycolysis Assay (ab197244; Abcam, Cambridge, UK) kit as per the manufacturer’s guidelines. Blank controls and experimental samples were each set up in triplicate. Serum from the model mice was collected, and levels of liver enzymes alanine aminotransferase (ALT) and aspartate aminotransferase (AST) were quantified colorimetrically.

### 2.5. Measurement of ATP Content, Mitochondrial Membrane Potential (ΔΨm), Intracellular ROS, Intracellular Ca^2+^ Concentration and Mitochondrial Permeability Transition Pore (MPTP) Opening

Cellular ATP production was quantified with an ATP bioluminescent assay kit (Beyotime, China). Following treatment with lysis buffer, the chemiluminescence per microgram of protein in sanguinarine-treated cells was compared to non-treated cells. ATP content was determined following standard protocols. Additionally, mitochondrial membrane potential (MMP), intracellular reactive oxygen species, Ca^2+^ concentration, and MPTP opening were assessed as previously outlined [30,31].

### 2.6. Cell Death and Cell Cycle Assay

Cells were plated into 12-well plates and treated with z-vad-fmk, necrostatin-1, or varying doses of sanguinarine. Subsequently, cells were stained with FITC-labeled annexin-V and PI as per the provided instructions [32]. Cell death was assessed using flow cytometric analysis (BD Biosciences, Franklin Lakes, NJ, USA). For the cell cycle analysis, samples were collected, fixed in 75% ethanol at −20 °C overnight, and then resuspended in staining buffer containing 20 mg/mL PI and 50 mg/mL RNase A. The PI-stained DNA was quantified using flow cytometry.

### 2.7. RNA Sequencing

The total RNA from the cell sample was extracted and utilized for library preparation. Subsequently, the libraries underwent sequencing on an Illumina HiSeq2000 platform. The sequencing data underwent quality control analysis using FASTQC (http://www.bioinformatics.babraham.ac.uk/projects/fastqc/ accessed on 6 May 2023), followed by differential gene expression analysis using the DESeq2 package (https://bioconductor.org/packages/release/bioc/html/DESeq2.html accessed on 28 April 2023) in the R platform. Gene Ontology (GO) analysis was performed for functional annotation of the differential genes, while pathway analysis was conducted to identify significant pathways based on the KEGG database (http://www.genome.jp/kegg/ accessed on 12 March 2023).

### 2.8. Quantitative Reverse Transcription-Polymerase Chain Reaction (qRT-PCR) and Western Blotting

The relative expression of each gene compared to GAPDH was calculated using the 2^∆∆CT^ algorithm. Primer sequences used in this study can be found in Table 1. Total protein, mitochondrial protein, and nuclear-cytosol protein were extracted following the provided instructions. Equal amounts of protein samples were separated on sodium dodecyl sulfate-polyacrylamide gel and transferred to polyvinylidene difluoride membranes. Signals were visualized using the Odyssey Two-color Infrared Laser Imaging System (Li-Cor, Lincoln, NE, USA). Details of the primary antibodies employed in this study are outlined in Table 2. Bands were quantified densitometrically in ImageJ (Image J Software, 1.8.0 National Institutes of Health, Bethesda, MD, USA), and full-length original western blots along with statistical graphs are included in the Appendix A.

### 2.9. Immunofluorescence Staining

Cell samples were fixed using 4% paraformaldehyde and permeabilized with 0.1% Triton X-100, then subjected to primary antibodies overnight; followed by appropriate secondary antibodies sequentially. The colon tissues were embedded in paraffin, and 4 μm sections were cut. They were incubated with indicated primary antibodies followed by secondary antibodies for immunofluorescence staining. Slides were incubated with DAPI solution and were observed under a fluorescence microscope (IX53, Olympus, Tokyo, Japan).

### 2.10. Clinical Tissue Array

IHC was implied to detect PKM2 expression in 53 paraffin-embedded tissues which were clinically and histologically diagnosed liver cancer. PKM2 antibody was used and rabbit anti-mouse IgG was used as the negative control antibody.

### 2.11. Animal Experiment

All protocols and animal studies were performed in accordance with the Guide for the Care and Use of Laboratory Animals by the US National Institutes of Health (NIH Publication No. 85-23, updated 2011). Animal use and all experiments involving animals were approved by the Ethical Committee of Tongji University. C57 male mice were obtained from the Shanghai Slack Laboratory Animal Co. Ltd. (Shanghai, China). Rosa26^RFP^ mice were both purchased from Jackson Lab (Bar Harbor, Hancock County, ME, USA). Alb-Cre^ERT2^ mice were gifts from the Institute of Biochemistry and Cell Biology, Shanghai Institutes for Biological Sciences, Chinese Academy of Sciences. PKM2^loxP/loxP^ mice were purchased from Nanjing Animal Model Institute (Nanjing, China). Cre^ERT2^-mediated recombination was induced by administrating tamoxifen (100 mg/kg, dissolved in 150 μL corn oil, daily for a week). PKM2 was knocked out in hepatocytes by constructing Alb-Cre^ERT2^; PKM2^loxP/loxP^; Rosa26^RFP^ mice, which received sanguinarine treatment with Alb-Cre^ERT2^; Rosa26^RFP^ control group was in parallel. HCC animal model was constructed using DEN solution (Sigma, St. Louis, MO, USA) at a dose of 30 mg/kg weight and 25% tetrachloromethane (CCl4). Sanguinarine treatment were administered 4 weeks after the first CCl4 injection. Mice were anesthetized and sacrificed 16 weeks after modeling, and tissue samples were collected for further analyses.

All the animals were free to consume food and water and kept in a colony room under conditions of constant temperature (25 °C), humidity (70%), and lighting (12 h light/12 h dark cycle) in the Tongji University Animal Center (Shanghai, China).

### 2.12. Statistical Analysis

Statistical analyses were carried out using the GraphPad Prism 8.0 software (GraphPad Software, San Diego, CA, USA). Data are presented as the mean ± standard deviation (SD) from three independent experiments. Flow cytometry results were analyzed using the FlowJo program (FlowJo, V10, LLC, Ashland, OR, USA). ModFit LT software (version 3.3) was applied to analyze the cell cycle data. Two-group comparisons were analyzed by Student’s *t* test, and multigroup comparisons were analyzed via one-way ANOVA. Spearman’s correlation analysis was performed to evaluate expression correlation. *p* < 0.05 was considered statistically significant.

## 3. Results

### 3.1. Sanguinarine Inhibited Proliferation, Metastasis, and Induced Cell Death of HCC Cells

Sanguinarine exhibited potent anti-hepatocellular carcinoma (HCC) activity in vitro. IC50 values of sanguinarine against Bel7402, HepG2, HCCLM3, and SMMC7721 were 2.90 μM, 2.50 μM, 5.10 μM, and 9.23 μM, respectively. The IC50 values of sanguinarine for different HCC cell lines were listed in the Table 3. Bel7402 and HepG2 displayed higher sensitivity to sanguinarine (Figure 1A). The cytotoxicity effects were concentration-dependent over the range of 2–4 μM; thus, in subsequent work, we used sanguinarine at 2 and 4 μM. In addition, drug treatment significantly inhibited the colony formation of Bel7402 and HepG2 cell lines (Figure 1B). TEM images revealed characteristic ultrastructural changes in treated tumor cells, such as cell membrane rupture, cytoplasm vacuolization, and damaged mitochondria; along with increased mitochondrial swelling, cristae disorder, and matrix particles (Figure 1C). To elucidate the mechanism of sanguinarine-induced cell death, Annexin V/PI staining was performed on treated HCC cells for flow cytometry analysis. A substantial proportion of cells clustered in the Q1 quadrant, and this subpopulation was predominantly rescued by necrostatin-1, but not affected by z-vad-fmk; suggesting that apoptosis played a minor role in sanguinarine-induced cell death (Figure 1D). Consistent results were obtained from the CCK8 assay (Figure 1E). Furthermore, western blot analysis revealed a significant increase in phosphorylated RIP3 (pRIP3) and phosphorylated MLKL (pMLKL) levels upon sanguinarine treatment, indicating activation of necroptosis (Figure 1F). Collectively, these findings underscore the anti-HCC effects of sanguinarine through the induction of necroptosis.

### 3.2. Sanguinarine Targeted PKM2 and Inhibited the Pyruvate Kinase Activity

RNA-sequencing was performed on treated cells to identify potential targets of sanguinarine. The GO analysis indicated that differentially expressed genes (DEGs) were primarily associated with lipid metabolism, energy metabolism, and cell growth and death pathways (Figure 2A). Therefore, we next detected the expression levels of glucose and lipid metabolism-related genes. PKM2 was screened as a significant target of sanguinarine using real-time PCR analysis (Figure 2B). Furthermore, pyruvate kinase activity in sanguinarine-treated cells exhibited a dose-dependent decrease (Figure 2C). Molecular docking analysis revealed a direct interaction between sanguinarine (CHEBI ID: 18319) and PKM2 (PDB ID: 1T5A), showing favorable binding affinity with key residues such as ARG436, HIS439, LYS266, and THR459 (Figure 2D). Additionally, molecular dynamics simulations were conducted to validate the stability of the sanguinarine-PKM2 molecular complex (Figure 2E).

### 3.3. Sanguinarine Reduced the Aerobic Glycolysis Level in HCC Cell Lines

To assess the impact of sanguinarine on glycolysis, the lactate content in cell culture media was measured. The findings indicated a significant decrease in lactate production in the high-dose group compared to controls within the same incubation period (Figure 3A). Additionally, there were notable differences in glucose uptake (Figure 3B). The ECAR graphs effectively demonstrated that the high-dose group exhibited lower glycolytic rates compared to the control group; highlighting the effect of sanguinarine on cellular glycolytic flux (Figure 3C). Moreover, the protein levels of key glycolytic enzymes were analyzed, revealing reduced expressions of HK2 and PKM2 in the high-dose group (Figure 3D). These results suggest that high-dose sanguinarine treatment suppressed aerobic glycolysis, particularly inhibiting the activity of PKM2 molecules.

### 3.4. Sanguinarine Inhibited Mitochondrial Bioenergetics

The mitochondrial structure was observed to be disrupted in the group treated with sanguinarine, as shown by electron microscopy. An increase in mitochondrial ROS production was identified as a key event in cell death. The current study revealed that the high-dose group exhibited elevated intracellular ROS production, MMP depolarization, Ca^2+^ concentration, and excessive opening of MPTP; as determined by cell flow analysis (Figure 4A–D). Assessment of oxidative phosphorylation and UCP2, crucial indicators of mitochondrial function through western blotting analysis indicated that treatment with 4 μM sanguinarine led to a reduction in the levels of OXPHOS subunits (Figure 4E). Additionally, quantitative PCR data demonstrated a decrease in mtDNA content and an increase in transcript levels of UCP2 in both cell lines (Figure 4F). Moreover, sanguinarine treatment was found to disrupt mitochondrial respiratory capacity, as evidenced by a decrease in cell OCR capacity and ATP content (Figure 4G,H).

### 3.5. Sanguinarine Exerted Anti-Tumor Effect by Regulating the PKM2/β-Catenin Axis

PKM2 functions as a transcriptional cofactor that phosphorylates β-catenin to facilitate its transactivation and involvement in tumorigenesis. Following sanguinarine treatment, a dose-dependent decrease in PKM2 nuclear translocation and β-catenin expression was observed through immunofluorescence staining (Figure 5A). The levels of total PKM2, nuclear PKM2, and nuclear β-catenin Y333 sites in cells were significantly reduced after a 6-h incubation with sanguinarine (Figure 5B). Sanguinarine treatment also impacted the distribution of cell cycle phases (Figure 5C) and led to decreased expression of cell cycle-related genes (Figure 5E). Furthermore, a ChIP assay revealed that sanguinarine inhibited β-catenin binding to the CCND1 promoter region; indicating its effect on the PKM2/β-catenin signaling pathway and downstream genes (Figure 5D).

### 3.6. Effect of Liver Specific PKM2 Deletion on HCC Progression In Vivo

To elucidate the clinical pathological significance of PKM2 in hepatocellular carcinoma (HCC) patients, we conducted a tissue microarray analysis of PKM2 in 53 primary tumor samples. Our results showed high expression of PKM2 in the cell nucleus in approximately 69.81% (37 out of 53) of cancerous samples (Figure 6A). Red fluorescent protein (RFP) was used to label hepatocytes (Figure 6B). Additionally, we developed an HCC mouse model by genetically engineering hepatocyte-specific PKM2 knockout Alb-CreERT2; PKM2loxp/loxp; Rosa26RFP mice and Alb-CreERT2; Rosa26RFP mice to investigate the impact of PKM2 depletion in vivo. Hematoxylin and eosin (H&E) staining revealed a larger area of heterogeneous cell masses and disrupted hepatic lobule structure in the liver of wild-type HCC animals, whereas neoplastic clusters and PKM2-positive tumor cells were reduced in the drug-treated group (Figure 6C). Malignant phenotypes were attenuated by either PKM2 knockdown or sanguinarine application. Liver tissues from PKM2 knockdown or sanguinarine-treated mice showed decreased tumor nodules, with no significant difference in spleen indices observed (Figure 6D). Furthermore, sanguinarine treatment led to increased protein levels of pRIP3 and pMLKL (Figure 6E). Immunohistochemistry (IHC) analysis demonstrated reduced levels of N-cadherin, vimentin, MMP2, and MMP9, along with increased levels of E-cadherin in liver tissues following sanguinarine intervention (Figure 6F).

## 4. Discussion

Hepatocellular carcinoma is a cancer with diverse metabolic characteristics, which often exhibit the Warburg effect due to oncogenic mutations that drive cell proliferation and increase energy requirements [33,34]. Despite this, the metabolic features do not always align completely with the classic Warburg effect patterns [3,35]. In our current study, we observed that cancer cells display heightened glycolysis, oxidative phosphorylation, and ATP synthesis to support their survival. Treatment with sanguinarine was found to reduce the glycolysis rate in HCC cells; as evidenced by a decrease in extracellular acidification rate (ECAR) and levels of the key glycolysis enzymes GLUT1, LDHA, and PKM2. Additionally, biochemical experiments showed that sanguinarine disrupted mitochondrial function and bioenergetics, leading to a decreased oxygen consumption rate (OCR). Our findings suggest that, by disrupting the metabolic processes of glycolysis and mitochondrial energy production, cells become vulnerable to metabolic crisis and subsequent necroptosis following treatment with the agent.

Our findings are in line with previous research indicating that treatment with sanguinarine can induce various forms of cell death to exert anti-tumor effects [36,37]. Importantly, our study also demonstrated favorable safety profiles for in vivo application (Appendix A). It has been established that cellular energy levels play a crucial role in determining the type of cell death that occurs, with energy deficiency being a key factor in initiating necroptosis [38,39,40]. Through in vitro experiments using HCC cells treated with different concentrations of sanguinarine for varying durations, we observed a significant decrease in ATP generation with increasing sanguinarine concentration. ATP, the primary energy source for cells, is produced through two main pathways: oxidative phosphorylation (OXPHOS) in mitochondria and aerobic glycolysis in the cytosol [41,42]. Glycolysis, a less efficient ATP-producing pathway, can lead to rapid depletion of glycolytic reserves, resulting in decreased ATP levels and subsequent necroptosis or apoptosis [43,44]. For instance, administration of citrate can deplete ATP and enhance epigenetic therapy by favoring necroptosis over apoptosis [45]. Furthermore, mitochondria play a crucial role in determining cell fate by regulating energy production and cell death pathways. Epigenetic drugs can activate dehydrogenase in complex II, leading to excessive reactive oxygen species production and induction of tumor necroptosis in AML [3]. Our study delved into the metabolic implications of drug-induced energy depletion and demonstrated that sanguinarine application resulted in cell necroptosis.

The PKM2 molecule has been identified in both the cytosol and nucleus of cancer cells. It has been reported that PKM2 plays a key role in regulating both glycolysis and mitochondrial energy metabolism [46]. Specifically, as a glycolytic enzyme, cytosolic PKM2 catalyzes the dephosphorylation of phosphoenolpyruvate to pyruvate [47]. Our study revealed that the enzyme activity of PKM2 was inhibited by varying doses of sanguinarine. While PKM2 primarily functions in regulating glycolysis in the cytosol, it is also present in the mitochondria and nucleus. Evidence suggests that the activation of PKM2 is closely linked to mitochondrial metabolism and material synthesis [48,49]. The translocation of PKM2 to the mitochondrial outer membrane is crucial for maintaining mitochondrial function. Additionally, PKM2 can activate the transcription of HIF-1, PDK1, and BNIP3, leading to a decrease in TCA intermediates and OXPHOS [50]. Undoubtedly, PKM2 serves as a critical switch in glucose metabolism and mitochondrial OXPHOS in tumors.

Drug-target site modification is a crucial aspect of anti-cancer drug development. Research indicates that acetylation of the K433 residue is a distinctive covalent modification of PKM2 [51]. Specifically, acetylation of PKM2 at K433 has been shown to enhance PKM2-DDB2 binding, thereby impacting hepatocyte survival [52]. Various small-molecule compounds have been identified to bind to specific residues of PKM2, regulating PKM2-mediated glycolysis in tumor cells. For instance, the natural product micheliolide selectively activates PKM2 over PKM1 by binding to its conserved cysteine424 residue, promoting tetramer formation and influencing the translocation of PKM2 into the nucleus [53]. Oncometabolites such as SAICAR have been found to support the growth and invasiveness of oral cancer cells by targeting PKM2, with specific binding sites within the activator site of PKM2 identified as GLY321, ARG436, HIS439, LYS266, and TYR466 [54]. Furthermore, 4-oxo-2-nonenal (ONE)—a product of lipid peroxidation—has been shown to modify Cys424 and His439 on the PKM2 molecule in vitro, leading to a dose-dependent decrease in PKM2 activity in RKO cells [55]. Molecular docking analysis has revealed multiple binding sites of sanguinarine on PKM2, resulting in kinase inactivation, although the precise biological functions of these amino acid sites remain unclear.

In contrast to the active enzymatic PKM2 tetramer, the nuclear PKM2 dimer plays a key role in signal transduction within tumor cells [56]. When translocated, PKM2 interacts with Wnt signaling to regulate the activity of OCT4 and STAT3 transcription factors, ultimately impacting tumor cell proliferation [57]. Our current findings demonstrate that sanguinarine targets PKM2 at the molecular level, as revealed by transcriptome sequencing. GO and KEGG enrichment analyses indicate that sanguinarine primarily affects metabolic processes and cell growth and death signaling. Treatment with sanguinarine disrupts the intra-molecular interaction of the PKM2/β-catenin axis, leading to inhibition of proliferation and EMT in HCC cells. These findings suggest that targeting PKM2 may serve as a promising therapeutic strategy for HCC.

Additional limitations exist in our study. More clinical and experimental experiments are needed to further confirm this conclusion. Besides, adverse effects or target-off effects should be investigated before clinical application. Other potential targets, aside from PKM2, may also benefit from sanguinarine anti-tumor therapy. Moreover, the lack of clinical studies involving PKM2 intervention across different patient cohorts necessitates further investigation into whether PKM2 can truly benefit tumor patients.

## 5. Conclusions

In summary, the natural product sanguinarine demonstrates anti-tumor properties by promoting necroptosis in hepatocellular carcinoma (HCC). Our research presents strong evidence supporting the role of sanguinarine in triggering necroptosis in HCC cells, leading to energy depletion through disturbances in mitochondrial respiration and decreased glycolysis. Furthermore, sanguinarine acts on PKM2 to inhibit tumor cell proliferation and metastasis by modulating the PKM2/β-catenin signaling pathway.

## Figures and Tables

**Figure 1 cancers-16-02533-f001:**
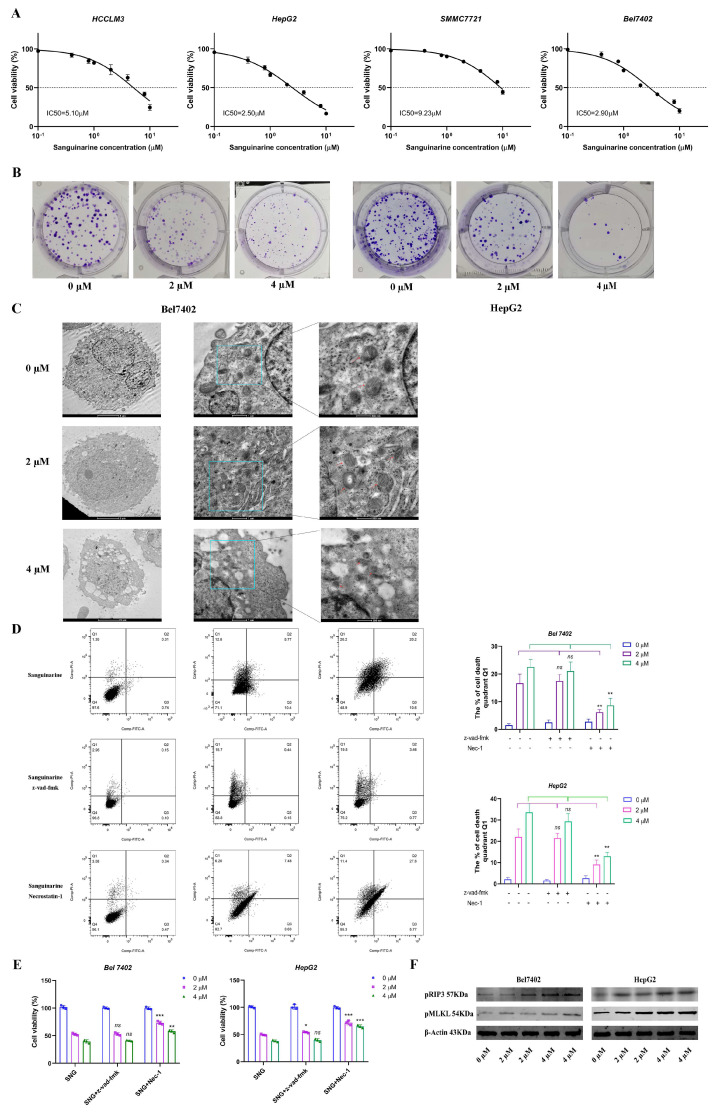
Sanguinarine inhibited proliferation and induced cell death of HCC cells. (**A**) Four HCC cell lines were incubated with escalating concentrations of sanguinarine for 6 h, and IC50 values were calculated. (**B**) The colony formation of HepG2 andBel7402 cells after treated with different doses of sanguinarine. (**C**) Morphological changes of HCC cells incubated with 2 μM and 4 μM sanguinarine were observed under scanning transmission electron microscope. Red arrows indicate mitochondrial morphological changes after sanguinarine application. (**D**) Flow cytometry analysis of HCC cell death using Annexin V-FITC/PI double staining. Cells were treated with different concentrations of sanguinarine, caspase inhibitors Z-VAD-FMK and necrosis inhibitors necrostatin-1. (**E**) CCK8 assay for tumor cells co-incubated with sanguinarine and z-vad-fmk or necrostatin 1. (**F**) Western blot analysis of the expression levels of necroptosis critical proteins pRIP3, pMLKL. Data are expressed as mean ± SD. * *p* < 0.05, ** *p* < 0.01, *** *p* < 0.001. Statistical significance was assessed using a two-tailed unpaired student *t* test. Original western blots are presented in Appendix A.

**Figure 2 cancers-16-02533-f002:**
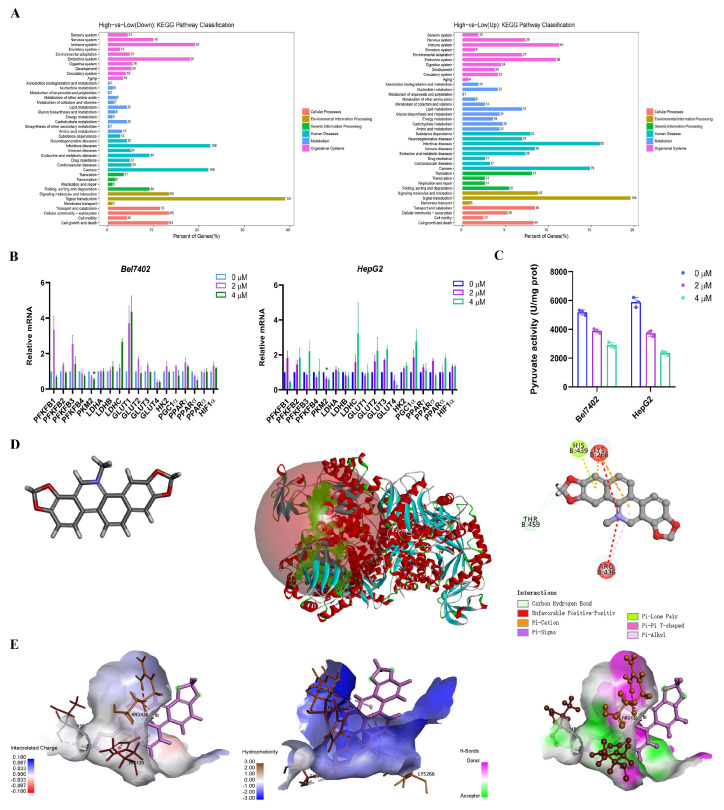
Sanguinarine targeted PKM2 and inhibited the pyruvate kinase activity. (**A**) Cell samples treated with 0 μM, 2 μM, 4 μM sanguinarine were applied for sequencing (*n* = 3). GO biofunction enrichment analysis of differentially expressed genes was shown. (**B**) Relative expression of lipogenesis and glucose metabolism genes in sanguinarine treated tumor cells. (**C**) Pyruvate kinase activity of two cell lines treated with sanguinarine were evaluated. (**D**) Three-dimensional structure of sanguinarine (CID: 5154). Three-dimensional view of the interaction between sanguinarine and PKM2 (PDB ID: 1T5A) with binding residues, bond distances, and types of bonds. Two-dimensional image of docked molecular structure between sanguinarine and PKM2 derived from Discovery Studio Visualizer. (**E**) Interactions between sanguinarine and PKM2: interpolated charge; hydrophobicity; H-bonds. Data are expressed as mean ± SD. * *p* < 0.05, ** *p* < 0.01, *** *p* < 0.001. Statistical significance was assessed using a two-tailed unpaired Student’s *t* test.

**Figure 3 cancers-16-02533-f003:**
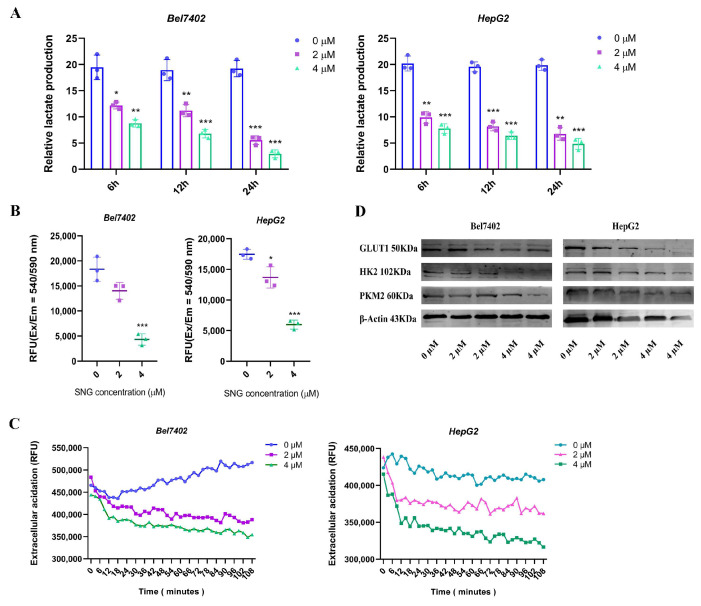
Sanguinarine reduced the aerobic glycolysis level of HCC cell lines. (**A**) Relative lactate production in different doses of sanguinarine treated HCC cells. (**B**) Cellular 2-DG uptake level. (**C**) Cellular glycolytic flux analysis. (**D**) Protein expression of glycolysis related genes were determined by western blot. Data are expressed as mean ± SD. * *p* < 0.05, ** *p* < 0.01, *** *p* < 0.001. Statistical significance was assessed using a two-tailed unpaired Student’s *t* test. Original western blots are presented in Appendix A.

**Figure 4 cancers-16-02533-f004:**
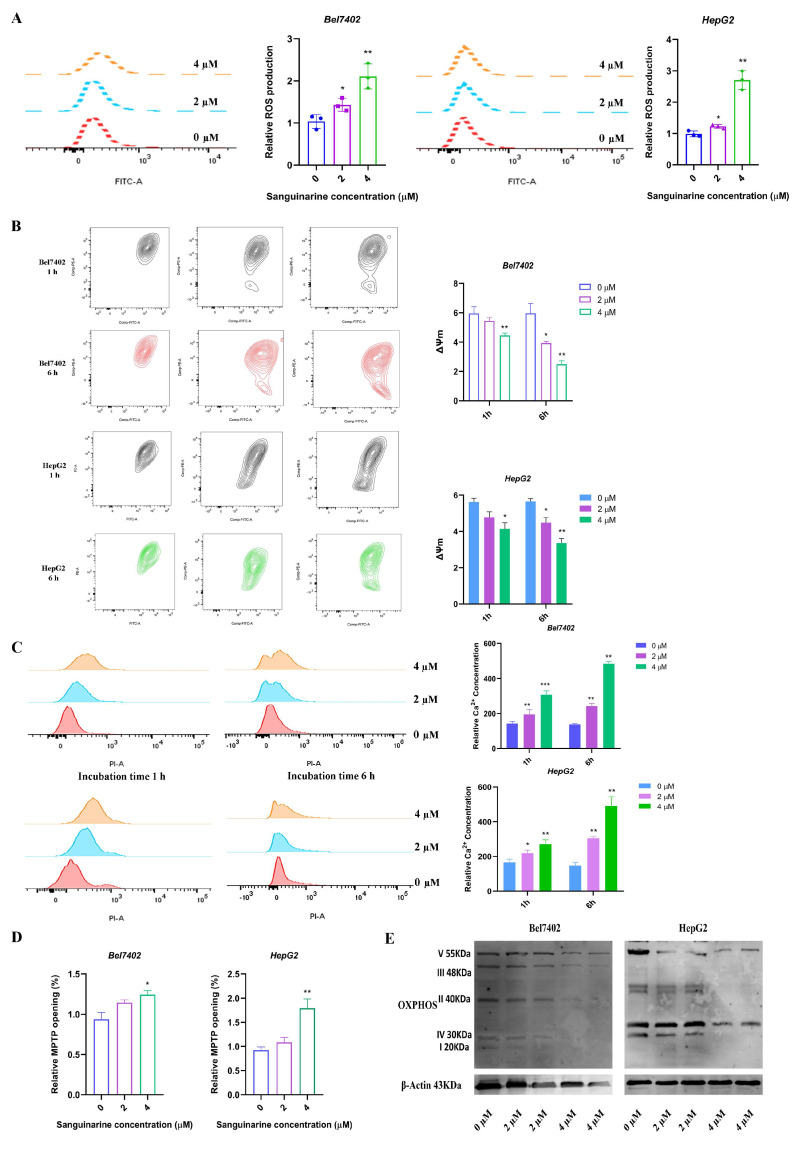
Sanguinarine inhibited mitochondrial bioenergetics. (**A**) Cellular ROS content in different concentrations of sanguinarine treated HepG2 and Bel7402 cells. (**B**) Mitochondrial membrane potential (MMP) analyzed by flow cytometry. (**C**) Intracellular Ca2+ content. (**D**) The change of mitochondrial mPTP opening via flow cytometry. (**E**) Representative western blots of OXPHOS and UCP2 in sanguinarine treated tumor cells. (**F**) Relative expressions of mitochondira bioenergetic related genes in sanguinarine treated cancer cells. (**G**) Oxygen consumption analysis using OCR assay kits. (**H**) Cellular ATP production of tumor cells exposed to different concentrations of sanguinarine. Data are expressed as mean ± SD. * *p* < 0.05, ** *p* < 0.01, *** *p* < 0.001. Statistical significance was assessed using a two-tailed unpaired Student’s *t* test. Original western blots are presented in Appendix A.

**Figure 5 cancers-16-02533-f005:**
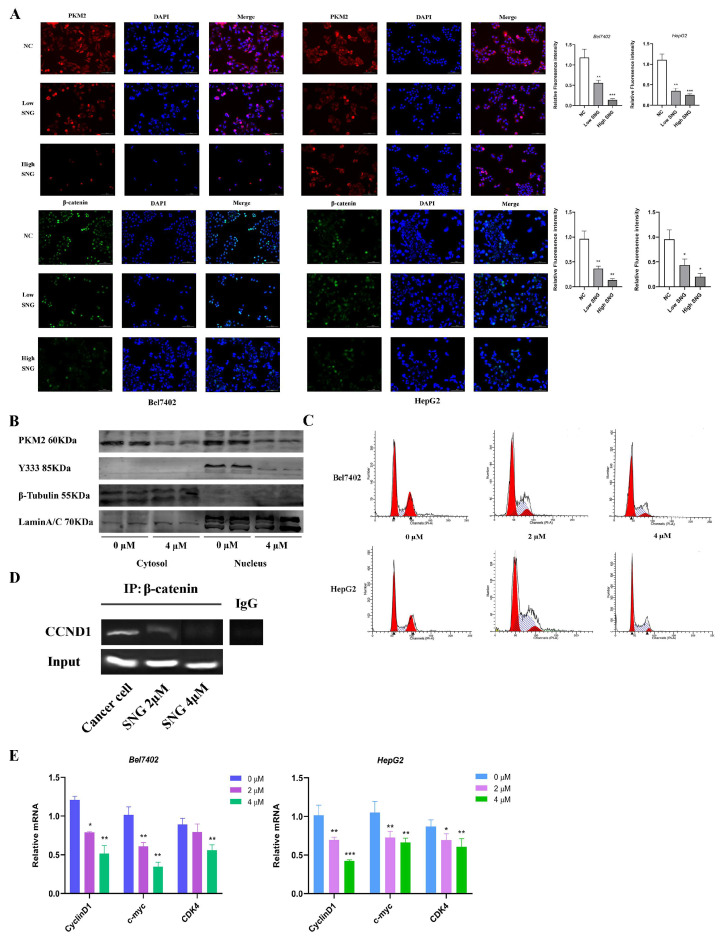
Sanguinarine exerted anti-tumor effect by regulating the PKM2/β-catenin axis. (**A**) Immunofluorescence staining of β-catenin and PKM2 in sanguinarine treated tumor cells, scale bar = 5 μm. (**B**) Western blot analysis of protein levels of PKM2 and β-catenin Y333 in cellular nuclear and cytoplasm, respectively. LaminA/C and β-tubulin were used as the internal references. (**C**) Effects of sanguinarine treatment on cell cycle distribution of tumor cells. (**D**) Co-IP assay was performed to determine the interaction between β-catenin and CCND1. (**E**) CyclinD1, c-myc, CDK4 RNA levels were measured by quantitative real-time RT-PCR. Data are expressed as mean ± SD. * *p* < 0.05, ** *p* < 0.01, *** *p* < 0.001. Statistical significance was assessed using a two-tailed unpaired Student’s *t* test. Original western blots are presented in Appendix A.

**Figure 6 cancers-16-02533-f006:**
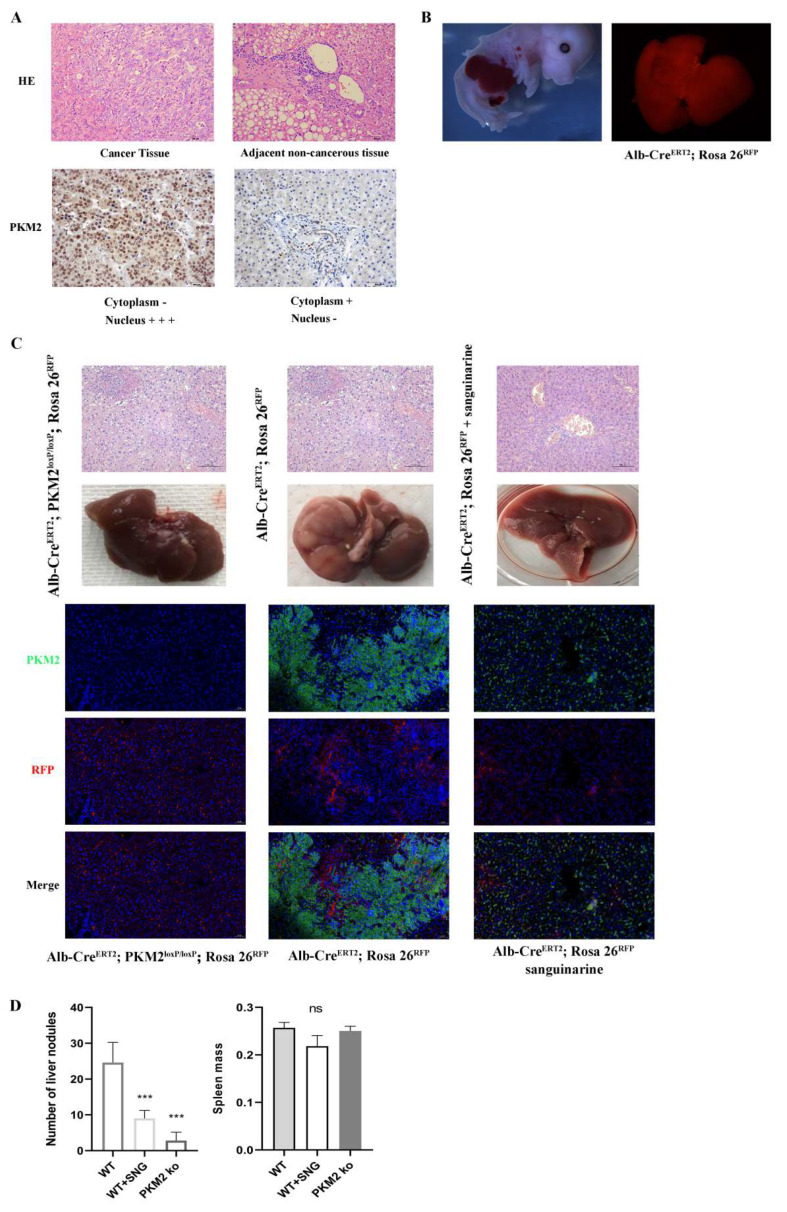
Effect of liver specific PKM2 deletion on HCC progression in vivo. (**A**) HCC specimens and paired adjacent normal tissues were stained with H&E. Immunohistochemical was performed to determine cytosolic and nuclear staining intensity of PKM2 in HCC samples. (**B**) Representative immunostaining pictures of Alb (RFP) in fetal liver. (**C**) Paired studies showed that PKM2−KO mouse and sanguinarine treated WT mouse displayed less tumor mass. Immunofluorescence staining for PKM2 (green) was performed. Rosa26^RFP^ reporter expression shows RFP-labelled hepatocytes (red). (**D**) The number of nodules on mice liver surface. The spleen mass of three groups. (**E**) Western blotting analysis of pRIP3, PKM2 and pMLKL in three groups. (**F**) IHC for PKM2/β−catenin axis, pRIP3, Ki67 and critical metastasis genes of liver tissues (200×). Data were expressed as mean ± SD. * *p* < 0.05, ** *p* < 0.01, *** *p* < 0.001, ns: no statistical significance. Scale bar = 100 μm. Statistical significance was assessed using a two-tailed unpaired Student *t* test. Original western blots are presented in Appendix A.

**Table 1 cancers-16-02533-t001:** Sequences of primer pairs used for amplification of mRNA by real-time PCR.

Genes	Sequencing
GAPDH	F TGTGGGCATCAATGGATTTGG
	R ACACCATGTATTCCGGGTCAAT
HK2	F GAGCCACCACTCACCCTACT
	R CCAGGCATTCGGCAATGTG
PFKFB1	F AGAAGGGGCTCATCCATACCC
	R CTCTCGTCGATACTGGCCTAA
PFKFB2	F TGGGCCTCCTACATGACCAA
	R CAGTTGAGGTAGCGTGTTAGTTT
PFKFB3	F TTGGCGTCCCCACAAAAGT
	R AGTTGTAGGAGCTGTACTGCTT
PFKFB4	F TCCCCACGGGAATTGACAC
	R GGGCACACCAATCCAGTTCA
LDHA	F ATGGCAACTCTAAAGGATCAGC
	R CCAACCCCAACAACTGTAATCT
LDHB	F TGGTATGGCGTGTGCTATCAG
	R TTGGCGGTCACAGAATAATCTTT
LDHC	F AGAACATGGTGATTCTAGTGTGC
	R ACAGTCCAATAGCCCAAGAGG
GLUT1	F GGCCAAGAGTGTGCTAAAGAA
	R ACAGCGTTGATGCCAGACAG
GLUT2	F TGTGGGCATCAATGGATTTGG
	R ACACCATGTATTCCGGGTCAAT
GLUT3	F GAGCCACCACTCACCCTACT
	R CCAGGCATTCGGCAATGTG
GLUT4	F GCTGCTCAACTAATCACCATGC
	R TGGTCCCAATTTTGAAAACCCC
PKM2	F ATGTCGAAGCCCCATAGTGAA
	R TGGGTGGTGAATCAATGTCCA
UCP2	F CCCCGAAGCCTCTACAATGG
	R CTGAGCTTGGAATCGGACCTT
SSBP1	F TGAGTCCGAAACAACTACCAGT
	R CCTGATCGCCACATCTCATTAG
MFN2	F CTCTCGATGCAACTCTATCGTC
	R TCCTGTACGTGTCTTCAAGGAA
OPA1	F TGTGAGGTCTGCCAGTCTTTA
	R TGTCCTTAATTGGGGTCGTTG
major arch	F CTGTTCCCCAACCTTTTCCT
	R CCATGATTGTGAGGGGTAGG
minor arch	F CTAAATAGCCCACACGTTCCC
	R AGAGCTCCCGTGAGTGGTTA
Cox II	F CCCCACATTAGGCTTAAAAACAGAT
	R ACCGCTACACGACCGGGGGTATA
PCG1α	F TGACACAACACGGACAGAAC
	R GCATCACAGGTATAACGGTAGG
β2M	F GCTGGGTAGCTCTAAACAATGTATTCA
	R CCATGTACTAACAAATGTCTAAAATGG
CCND1	F TTTTAGGGTTACCCCCTTGG
	R GCAAAGAATCTCAGCGAC
Cyclin D1	F GCTGCGAAGTGGAAACCATC
	R CCTCCTTCTGCACACATTTGAA
c-myc	F GGCTCCTGGCAAAAGGTCA
	R CTGCGTAGTTGTGCTGATGT
CDK4	F GGGGACCTAGAGCAACTTACT
	R CAGCGCAGTCCTTCCAAAT
PPARα	F ATGGTGGACACGGAAAGCC
	R CGATGGATTGCGAAATCTCTTGG
PPARγ	F GATGCCAGCGACTTTGACTC
	R ACCCACGTCATCTTCAGGGA
PPARδ	F CAGGGCTGACTGCAAACGA
	R CTGCCACAATGTCTCGATGTC
HIFα	F GAACGTCGAAAAGAAAAGTCTCG
	R CCTTATCAAGATGCGAACTCACA

**Table 2 cancers-16-02533-t002:** Antibodies.

Antibody	Supplier
β-actin	Cell Signaling Technology
pRIP1	Cell Signaling Technology
pRIP3	Cell Signaling Technology
pMLKL	Cell Signaling Technology
GLUT1	Proteintech
PKM2	Proteintech
HK2	Proteintech
β-catenin (phospho Y333)	Abcam
β-Tubulin	Abmart
LaminA/C	Proteintech
OXPHOS	MitoSciences

**Table 3 cancers-16-02533-t003:** IC50 values of sanguinarine for different HCC cell lines.

Cell Lines	IC50 Values
HCC LM3	5.10 μM
HepG2	2.50 μM
SMMC7721	9.23 μM
Bel7402	2.90 μM

## Data Availability

The datasets used and/or analyzed during the current study are available from the corresponding author on reasonable request.

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
