# Peer review of "Sanguinarine Induces Necroptosis of HCC by Targeting PKM2 Mediated Energy Metabolism"

_cancers, 2024, doi:10.3390/cancers16142533_

Round 1
Reviewer 1 Report
Comments and Suggestions for Authors
This study demonstrates Sanguinarine, a plant-derived product, potentially could be a promising drug for treatment of patients with liver disease. The most common cause of cirrhosis and hepatocellular carcinoma worldwide is quickly becoming steatohepatitis, both alcoholic and non-alcoholic. Have you or your colleagues in Shanghai studied these patients to assess the efficacy of Sanguinarine in these patients in an attempt to reduce the incidence of HCC?
Reviewer 2 Report
Comments and Suggestions for Authors
The manuscript “Sanguinarine Induces Necroptosis of HCC by Targeting PKM2 Mediated Energy Metabolism” investigates the effects of sanguinarine on hepatocellular carcinoma (HCC) cells, focusing on its impact on energy metabolism and necroptosis via targeting the PKM2/β-catenin axis.
Major Comments
1. The introduction provides a good overview of the metabolic alterations in HCC and the potential therapeutic role of sanguinarine. However, it could benefit from a more detailed discussion on the current state of necroptosis research in cancer therapy to better contextualize the study.
2. The manuscript mentions using different concentrations of sanguinarine but lacks proper justification for the chosen doses. It would be beneficial to include preliminary dose-response data or literature references to support these choices.
3. The ethical considerations and the justification for the use of animal models need more elaboration. Details about the number of animals used, randomization, and blinding during the experiments should be mentioned clearly.
4. The IC50 values are provided, but there is a need for a statistical comparison between the different cell lines to highlight the sensitivity variations. Additionally, the figures showing cell viability and colony formation need better labelling and resolution for clarity.
5. The images provided in the TEM section are not very clear. Higher resolution images with detailed legends describing the observed morphological changes are necessary.
6. The RNA-seq data is summarized, but there is insufficient detail on the selection criteria for differentially expressed genes. Validation of RNA-seq results using qRT-PCR for key genes should be included, along with corresponding statistical analyses.
7. The discussion section needs to critically evaluate the results in the context of existing literature. There should be a deeper analysis of how sanguinarine’s mechanism compares with other known HCC therapies.
8. Potential limitations of the study, such as the specificity of sanguinarine for PKM2 and any off-target effects, should be addressed. Additionally, future directions for research should be discussed.
9. Figures need to be more descriptive, with comprehensive legends explaining all aspects of the data presented. The resolution of figures, especially those depicting microscopy images, should be improved.
10. Tables summarizing key quantitative findings, such as cell viability percentages, IC50 values, and gene expression changes, should be included for easier reference.
Minor Comments
1. Ensure all references are up-to-date and relevant. There are a few instances where recent studies could provide additional support or context to the findings.
2. The supplementary tables mentioned (Supplementary Table 1 for primer sequences and Supplementary Table 2 for antibodies) should be included within the manuscript or as an appendix for ease of access.
Comments on the Quality of English LanguageThere are several grammatical errors and awkward phrasings throughout the manuscript. A thorough proofreading is necessary to improve readability. Consistency in terminology (e.g., “sanguinarine” vs. “Sanguinarine”) should be maintained.
Reviewer 3 Report
Comments and Suggestions for Authors
Dear Author,
The research article titled Sanguinarine induces necroptosis of HCC by targeting PKM2 mediated energy metabolism by Rui Kong et al was well described.
It is know well know that sanguinarine is a bioactive benzophenanthridine alkaloid isolated from the roots of Sanguinaria canadensis, commonly known as bloodroot. This compound is known for its potential medicinal properties, including anti-inflammatory, antimicrobial, and anticancer effects. Sanguinarine can interact with cellular components, leading to apoptosis (programmed cell death) in cancer cells. In this study authors demonstrated that Sanguinarine treatment induces necroptosis in liver cancers cells based on the western blot and TEM studies. Furthermore, the author explained the Warburg effect.
Overall, the article is interesting to read and their finding.
I would recommend this manuscript for next level with minor corrections.
Minor changes:
1. Figure 1C. the morphological changes of TEM images are not clear. It will be better if author can enlarge the specific area to show the mitochondrial changes. Also, author can use the positive control along with their compound.
2.Figure 1F, the western blot study is not very clear. I could not make any difference in pRIP3 in HepG2. In parallel, author can show the quantification of protein bands and its significance.
3. Figure 5A, the immunofluorescence data is not very clear for both PKM2 and beta-catenin.
Round 2
Reviewer 2 Report
Comments and Suggestions for Authors
Thanks for addressing the raised concerns.